# Multi-Component Interventions in Older Adults Having Subjective Cognitive Decline (SCD)—A Review Article

**DOI:** 10.3390/geriatrics8010004

**Published:** 2022-12-27

**Authors:** Madhuchhanda Mohanty, Prakash Kumar

**Affiliations:** 1Department of Occupational Therapy, Amity University, Noida 201301, India; 2Department of Occupational Therapy, Pandit Deendayal Upadhyaya National Institute for Persons with Physical Disabilities, New Delhi 110002, India

**Keywords:** subjective cognitive decline, Alzheimer’s disease, ageing disorder, cognition, multicomponent nonpharmacological intervention

## Abstract

Subjective cognitive decline (SCD) is one of those significant concerns faced by older individuals. Though it is predominantly self-reported, it is not an event that should be overlooked, considering its significant association with cognitive disorders like Alzheimer’s disease, mild cognitive impairment, and so on. This makes it imperative to find ways to manage the event to enhance the cognitive performance of older adults and/or suppress the rate at which cognitive decline results in impairment. While multiple interventions have been used for SCD, multi-component non-pharmacological interventions are beginning to gain more attention among researchers. This is due to how such interventions have effectively contributed to improved cognitive performance across different outcome domains. Against this backdrop, this literature review has been conducted to explore the different multi-component non-pharmacological interventions utilized in managing SCD. Papers from databases such as PubMed, Scopus, and EBSCO were retrieved, with relevant data being extracted on the subject matter to address the objective of this review.

## 1. Introduction

Elderly population is increasing rapidly worldwide, and so is the number of older adults having age-related cognitive decline, as well as dementia. Ageing is associated with a decline in cognitive functions that are critical to independence, social engagement, and quality of life. This decline in cognitive function, if not checked, can advance to clinical cognitive decline and in turn can progress to dementia [1,2,3]. As the proportion of people aged over 65 years continues to expand, it is estimated that, by 2050, dementia could affect some 106.2 million people globally [4]. Age-related cognitive decline affects far more people than dementia [5]. Most dementias are caused by Alzheimer’s disease (AD), and there is no cure available for AD or other types of dementia [6]. Therefore, it is important to develop an intervention to reduce the number of dementias. It has been discovered that pathophysiology associated with AD starts ten years or more before objective cognitive impairment that may be assessed using standardized neuropsychological instruments [6,7]. The seven risk factors for dementia are diabetes, hypertension, obesity, smoking, depression, lower education, and physical inactivity. It is estimated that a 10–25% improvement in all seven risk factors could potentially prevent up to 1.1 to 3.0 million cases of AD worldwide [8]. Dementia prevention primarily focuses on reducing the risk factors and enhancing the lifestyle of the middle-aged population at very early stages before the onset of symptoms and secondarily by trying to reduce or halt the progression of a disease once the symptoms start to appear [8,9]. The potential early symptomatic manifestation of AD is subjective cognitive decline (SCD), a pre-clinical stage of AD. Researchers were motivated to turn their attention to the preclinical stage of AD when several earlier clinical trials of treatments in dementia or mild cognitive impairment (MCI) stages failed [10,11]. SCD is a cognitive state that lies between objective cognitive impairment and intact cognition [12].

Subjective cognitive decline (SCD) is the self-reported experience of worsening memory or cognition or more frequent confusion or memory loss. The distinction cannot be made by cognitive testing, because individuals with SCD and cognitively unimpaired individuals without SCD are, by definition, objectively unimpaired, and they both perform above the cut-off for impairment in cognitive tests [13]. Subjective cognitive decline (SCD) represents a significant concern among the ageing population, and it is more often than not self-reported, although it is also possible for an informant to report about another person having SCD based on specific observations [14]. Some of the most common symptoms of SCD are depression, anxiety, and cognitive complaints [14,15]. Furthermore, an individual with SCD may also experience a heightened level of stress, anger, and fear of dementia. The prevalence of SCD is around 18–55% globally [16], despite the unavailability of any concrete objective measure to ascertain the decline [17]. The prevalence figure comes even in the face of how the cognitive tests of the majority of the individuals’ nursing worries over SCD have often turned out within the normal (healthy) range [12]. Besides, it has been widely acknowledged that people with SCD have better access to cognitive reserve and also exhibit considerably preserved current cognitive function [18]. There is also a higher probability for older adults with SCD show neurodegeneration and other Alzheimer’s disease biomarkers than others without SCD [19,20,21,22]. As no disease-modifying treatment has been found to be effective in the treatment of cognitive impairment and Alzheimer’s disease, researchers are focusing on non-pharmacological treatment. The dementia research community has increasingly concentrated on multidomain therapies that address numerous risk variables at once in order to give a larger possibility of achieving observable improvements during study periods [23]. The emphasis is on close attention to the preclinical phases of cognitive disorders in a bid to ensure early detection that would further boost the chances of creating suitable prevention and interventions that adequately take care of such cognitive disorders [24].

### 1.1. Management of Subjective Cognitive Decline

The effective management of subjective cognitive decline is regarded as an incumbent need considering the rising elderly population across the globe, and the significant association between SCD and the heightened risks of pathological ageing [25]. Managing SCD does, however, come with specific challenges, as there have been contradicting outcomes regarding the effectiveness of certain non-pharmacological or even pharmacological interventions. The risk factors for SCD identified are older age, female sex, anemia, thyroid diseases, lack of physical exercises, living alone, minimal anxiety symptoms, and daytime dysfunction [26]. Interventions that could slow cognitive ageing or lower the risk of dementia are promised by the idea of cognitive reserve. Many recent studies have focused more intently on lifestyle characteristics, intellectually stimulating behaviors, and personality aspects, while the original observations focused on clearly measured variables like education or occupational accomplishment. Overall, they show that cognitive reserve is not a constant entity but can change throughout the course of a person’s lifespan depending on exposures and actions and that contributions to reserve originate from a variety of sources. It is encouraging to think that lifestyle adjustments made even later in life may provide protection against dementia or age-related cognitive impairment. Despite this intriguing possibility, careful research will be necessary to turn this concept into a workable solution. Such research would provide useful details both about the combination and timing of activities that may lead to maintaining or improving cognition and more successful ageing [18].

### 1.2. Multi-Component Intervention

It is believed that the probability of gaining more satisfactory outcomes on cognitive performance will be boosted as multitask activities are engaged [27]. Furthermore, there has been evidence that multi-component non-pharmacological interventions are more effective than single-component interventions, since the former will bring about positive impacts in more than one domain [28]. That said, the perceived economic advantages offered by non-pharmacological interventions for the management of SCD makes people see them as a good alternative, even though studies in this direction are hard to come by [29]. A review reported that multi-domain lifestyle intervention strategies are effective in the delay and/or prevention of cognitive impairment in healthy older individuals [30]. Nevertheless, people should start performing them regularly as early as midlife, so that they could have an impact on cognitive function in later life. The results confirm that diet/nutrition, cognitive training, and physical exercise interventions are particularly effective in this sense [30]. This opinion is currently supported by the European Dementia Prevention Initiative, an investigator-initiated initiative of several groups involved in ongoing dementia prevention trials in Europe [31]. Since 2017, this initiative has been spreading to other continents and the reduplication of European projects e.g., Finnish Geriatric Intervention Study to Prevent Cognitive Impairment and Disability (FINGER), have been undertaken in Asia, USA, and Australia [32].

The benefits of physical activity for brain health have been linked to increased cognitive performance, lowered anxiety and depression risk, better sleep, and higher quality of life [33]. Physical activity may also be a promising intervention in the cognition of included patients with dementia and moderate cognitive impairment (MCI), according to recent meta-analyses of randomized controlled studies [34,35]. This evidence was also supported by cross-sectional investigations, long-term observational studies, and prospective intervention trials [36,37]. While a large number of studies have examined biomarker data to identify the underlying processes by which physical activity (PA) safeguards the health of the brain in healthy individuals and animal models, a 6-month resistance training program improved the activation of three critical cortical areas, i.e., the right frontal pole, the right occipital-fusiform gyrus, and the right lingial gyrus, during an associative memory test in a Canadian RCT involving 86 female participants with MCI [38]. An Australian SMART (Study of Mental Activity and Resistance Training) trial, involving 100 older people with MCI and six months of progressive resistance training, was linked to enhanced global cognition, a slowed progression of white matter lesions on MRI, and improved global cognition [39]. Finally, a 6-month aerobic PA intervention improved global cognitive function as measured by the ADAS-Cog, physical fitness as measured by the 6-min walking test, and diastolic blood pressure compared to usual care plus education in a Canadian RCT with 70 older adults with mild vascular cognitive impairment [40].

Cognitive training has become one of the most popular non-pharmacological interventions, primarily due to its effectiveness [41,42]. However, some studies have shown this particular intervention to bring about no improvement in cognitive performance [43]. Specifically stating, cognitive training has been reported to result in a significant increase in the volume of gray matter in the brain, even as cortical volume expansion was also achieved. The probability of SCD patients showing improvement in verbal recall increased [44]. Lifestyle interventions hovering around nutrition, exercises, and one or more component(s), such as behavioral modification, counseling, and so on [45], have also been utilized as a viable multi-component intervention, and this has reportedly caused the rate of cognitive decline to be slowed down, thus allowing elderly individuals to attain some degree of independence [46,47].

The Mediterranean diet is one dietary pattern that has shown promise in recent studies (MeDi). The MeDi is a diet that emphasizes a primarily plant-based diet with a high intake of fruits, vegetables, nuts, and legumes; a moderately high intake of fish; a low intake of red meat; and the primary source of fat being extra virgin olive oil [48]. The MeDi has been proven to directly reduce the risk of dementia through decreased levels of amyloid plaques [49], brain atrophy [50], and structural connections, as well as indirectly through changing cardiovascular risk factors and brain atrophy [51]

A substantial body of literature is developing that supports the notion that frequent social activity may help to prevent or delay cognitive decline in old age. Social activity could also provide meaningful social roles and a sense of purpose in old age [52], which could have direct neurohormonal influences on the brain, including the reduction of the stress response. Finally, although we controlled for physical activity, social activity also requires a degree of physical activity above and beyond regular exercise and walking, which could enhance cardiopulmonary fitness, leading to vascular changes in the brain and cerebral oxygenation that might protect against neuropathology [53].

It follows that, despite the advances made in finding a tenable treatment or management measure for SCD, there is yet to be a consensus as to the most suitable management protocol. Continued efforts are being directed at finding the most suitable interventions to effectively manage subjective cognitive decline. Researchers have gone ahead and employed single-component interventions, as well as multicomponent ones, but there still appear to be contrasting outcomes that are attainable from such measures. However, the focus on multicomponent interventions for SCD is yet to be extensively considered, and this is where this present study finds justification. In essence, the researcher will conduct a literature review to identify and explore papers reporting the use of multicomponent interventions in managing SCD. No review has been done on the use of multi-component non-pharmacological interventions for subjective cognitive decline. The objective of this study is to discuss available non-pharmacological multi-component interventions in the improvement and management of subjective cognitive decline in older adults. This information will enable researchers and clinicians to develop a suitable strategy for effectively managing or delaying cognitive impairment.

## 2. Materials and Methods

The methodology of the study, with reference to the search strategy, the selection of papers, and the inclusion and exclusion criteria, will be discussed in this section.

### 2.1. Sources of Information

In this study, the researcher gathered data from papers extracted from PubMed, Scopus, and EBSCO databases. Searches were run on each of these databases using the appropriate keywords and search strategy, which will be discussed later in this section. Utilizing two or more databases gave room for the retrieval of enough materials to address the objective of the study [54].

### 2.2. Years of Publication

As far as this review is concerned, the researcher limited the years of publications used to 2012–2022, meaning a timeframe of 10 years was ensured. This allows the researcher to explore more current data for the subject matter being addressed. As a result, the review’s findings can be better integrated into further studies aimed at arriving at more effective interventions for subjective cognitive decline.

### 2.3. Search Strategy

The databases were searched using the search terms highlighted as follow: ‘memory’ or ‘subjective cognitive decline’ or ‘subjective cognitive impairment’ or ‘subjective memory complaint’ or ‘subjective memory decline’ or ‘subjective cognitive complaint’ or ‘cognitive complaint’ AND ‘Multi-component interventions’ or ‘multidomain intervention’ or ‘multidomain lifestyle intervention’ or ‘training’ or ‘intervention’ or ‘therapy’ or ‘treatments’ or ‘non-pharmacological’ or ‘multiple training modalities’ AND ‘physical exercise’ or ‘physical activity’ or ‘physical therapy’ or ‘occupational therapy’ or ‘cognitive behaviour therapy’ or ‘cognitive training’ or ‘cognitive stimulation’ or ‘memory training’ or ‘mind body intervention’ or ‘computer games’ or ‘lifestyle’ or ‘socialisation’ or ‘nutrient’ or ‘nursing’ or ‘psychological’ or ‘psychosocial’ or ‘complementary therapies’ or ‘yoga’ or ‘spirituality’ or ‘activities of daily living’ AND ‘quality of life’.

### 2.4. Inclusion and Exclusion Criteria

The articles included were (1) randomized controlled trials (RCTs), cohort studies, reviews, and systematic reviews; (2) studies published in English only between 2012 and 2022 (May); (3) subjects aged 45 years or older, without gender and race restrictions; (4) subjects having self-experienced decline in cognitive capacity (unrelated to an acute event) despite normal performance on standardized cognitive tests, and failure to meet the criteria for MCI or dementia; (5) subjects from health care, memory clinics, or community settings; and (6) intervention strategies of all types of non-pharmacological Interventions (physical, psychosocial, cognitive, lifestyle, mindfulness, nutrition, etc.), including multicomponent interventions.

Exclusion criteria were (1) interventions for patients with mild cognitive impairment or Alzheimer’s disease; (2) subjects with objective impairment in neuropsychological tests; (3) subjects without cognitive complaints; (4) history of stroke, systemic diseases, other central nervous system diseases (e.g., Parkinson’s disease, tumors, encephalitis, and epilepsy), major depression, psychosis, and medical causes that may cause cognitive impairment; and (5) pharmacological intervention studies.

### 2.5. Selection of Papers

A total of 1845 papers were recovered after the search was run through all the databases that were resorted to. This number was trimmed down to 655 papers after reviewing the titles and removing duplicates. The papers were further reduced to 70 papers after abstracts were evaluated, and it was found that some of the papers were duplicates, while some did not meet the inclusion criteria—there were papers in other languages (asides English) and others addressing other concerns. The 70 papers were then reviewed, with the reviewer taking out those that were on MCI, Alzheimer’s diseases, non-RCT publications, non-cohort studies, and editorials. This led to the selection of 12 papers that were extensively reviewed after they had fulfilled the inclusion criteria (Figure 1).

## 3. Results

### 3.1. Assessment of Outcomes

The variables include psychological well-being, objective cognitive performance, and metacognition. The sway towards metacognition, which basically borders on how people perceive their cognitive performance (Fleming & Dolan) [55], was informed by the underestimation [of cognitive performance] that often prevails in most cases of SCD (Metternich, Schmidtke, and Hull cited in Bhome et al.) [16,56]. It has also been found that individuals with SCD often show relatively poor metacognitive performance, and as such, the effects of SCD may be reduced as metacognition is enhanced. The prevalence of distress in cases of SCD makes psychological well-being highly important to measure when attempting to establish whether or not an intervention is effective [14]. Again, objective cognitive performance can also not be missed, as it has widely been used in several studies bordering on the management and treatment of SCD [39], and its (objective cognitive performance) adoption will help to create a good balance in the evaluation of the effectiveness of SCD intervention(s).

### 3.2. Study Yield

All except two of the studies reviewed for this study were randomized controlled trial, and the other two studies were quasi-experimental and prospective controlled studies. The cumulative population for all 12 studies was 2687 subjects, with the highest (study population) having 1680 subjects while the lowest had 40 subjects. As per the characteristics of the multicomponent nonpharmacological SCD interventions, five papers (Chan et al.; Boa Sorte Silva et al.; Zuniga et al.; Ramnath et al.; Hong et al.) [25,57,58,59,60] mentioned interventions bordering on exercises; education (Chan et al.; Kwok et al.; Frankenmolen et al.; Cohen-Mansfield et al.) [25,41,61,62] and the use of computer programs (Ramnath et al.; Frankenmolen et al.; Pereira-Morales et al.; Oh et al.) [59,61,63,64] were mentioned in four studies. Multicomponent interventions on lifestyle modifications (Chan et al.; Hong et al.; Andrieu et al.) [25,60,65] and dietary measures (Hong et al.; Chan et al.) [25,60] were also adopted in three and two studies, respectively. The summary of the papers that were reviewed is shown in Table 1.

## 4. Discussion

The significance of multicomponent nonpharmacological interventions for managing subjective cognitive decline is well identified in some of the studies that were reviewed. The findings reported marked a reduction in subjective memory complaints by the participants [25,62,63,64,66]. This does not, however, necessarily mean that the interventions will ultimately lower the incidences of subjective memory impairment, which is more of a psychological problem [58,61]. As seen in one study, the multiple-modality intervention, along with mind-motor training, was found to significantly impact the subjects’ global cognitive functioning compared to the control group that also utilized similar multiple-modality intervention but without mind-motor training [57]. Likewise, the Dejian Mind–Body Intervention (DMBI) also resulted in significant improvements in immediate and delayed memory recall, as well as boosting the subjective feelings of physical and psychological health [25]. Again, the prospect of individuals having SCD being mindful of their state through educational courses and in turn improving their cognitive performances was also reflected in one study [62]. The study revealed how subjects improved in the aspects of executive function, attention, and visual–spatial function after taking courses that hover around different domains—memory, attention, cognition, and so on. The adoption and effectiveness of computer programs and/or games can also not be overlooked in the face of the positive impacts recorded in some of the studies wherein they were used.

Multiple-modality mind–motor training: As seen in one study, the multiple-modality intervention (involving aerobic exercise, resistance exercise, and stretching) along with mind–motor training (square-stepping exercise) was found to significantly impact the subjects’ global cognitive functioning compared to the control group that also utilized similar multiple-modality intervention but without mind–motor training. Both experimental and control groups showed slight improvements in concentration, reasoning, and planning [57]. The square-stepping exercise (SSE) [67] is a novel form of mind–motor training that has been associated with positive effects on global [68] and domain-specific cognitive functioning [68,69] in older adults.

Dejian Mind–Body Intervention (DMBI): Likewise, the Dejian Mind–Body Intervention (DMBI) also resulted in significant improvements in immediate and delayed memory recall, as well as boosting the subjective feelings of physical and psychological health. Dejian Mind–Body Intervention (DMBI) involves the Chan practice (whereby subjects were guided on self-awareness and self-control in relation to unrealistic goals); Nei Gong practice (that has to do with the teaching of mind–body exercises bordering on a series of breathing exercises and calm and gentle movements); and dietary modifications. Conventional Memory Intervention (CMI): Involves psychoeducation, mnemonic training, and progress monitoring [25].

Educative courses: The prospect of individuals having SCD being mindful of their state through educational courses and in turn improving their cognitive performances was also reflected in one study. Educative courses used were: the Health Promotion Course (HPC) involving teaching topics like health behaviors; dementia, and delirium; communication; cognitive activities to keep the mind fit; relationships; depression; life-long learning, etc; Cognitive Training Course (CTC): Hinges on the Advanced Cognitive Training for Independent and Vital Elderly (ACTIVE) course. It focuses on memory, reasoning, and processing speed, but only the memory training aspect was employed in this study; and Participation-centred Course, which focuses on memory and cognitive and organizational strategies. All interventions showed significant improvements in global cognitive function score compared to what was obtainable for the baseline. However, the CTC group was observed to have a significantly better global cognition score than the other groups. Slight improvements in executive function, visual spatial function, and attention were found across all the groups. Loneliness and self-report on memory difficulties also declined with all interventions [62].

Computer programs and games: The adoption and effectiveness of computer programs and/or games can also not be overlooked in the face of the positive impacts recorded in some of the studies wherein they were used. For example, the SMART and Fit Brains models employed by Oh et al. were reported to have significantly improved working memory scores and reduced subjective memory complaints. [64]. In Ramnath et al.’s study, interactive video games (IVG) showed significant improvements in cognitive measures of processing speed and executive functioning in the IVG group compared to the CMI group [59]. In another study, the Integrated Psychostimulation Program (IPP), a combination of computerized cognitive training (CCT) and conventional cognitive training (like meta-memory activities, progressive relaxation exercises, group discussions, and pen and paper exercises) was done for 90 min for 4 days per week [63]. The overall results indicate that the IPP was more effective in the improvement of cognitive performance and symptoms of anxiety compared with the group that only received sessions with the CCT, although cognitive training using technologies showed favorable results.

Memory strategy training (MST): in a study, the effect of a memory strategy training, including psychoeducation, cognitive restructuring, and strategy training, in older adults with SMC was examined. For the active control memory training, the computer program was used. It was concluded that memory strategy training improves memory functioning in daily life situations. The combination of psycho-education, expectancy management, sharing experiences in a group, and training to use memory strategies in daily life situations is effective in older adults with subjective memory complaints and could be implemented in clinical practice [61]

Physical Activity: Physical activity is the key to ascertaining the level of effectiveness of a nonpharmacological intervention. Five studies utilized physical activity in different forms, such as aerobic activities, resistive activities, stretching activities, standing and balancing, walking, coordinated stepping, etc. Physical activity coupled with mental activity (multi-modality mind–motor) has shown to produce significant improvement in cognition [57]. In contrast, in a study, the aerobic exercises and resistance training employed for 12 months did not yield any significant improvement in subjective memory impairment but significantly improved happiness levels [58]. Interactive video gaming involving standing and balancing activities has yielded positive results [59]. In a study, Chinese Chan-based lifestyle intervention was experimented on, involving some sort of mind–body exercises along with self-control training and dietary modifications to have improved memory functioning [25]. In another study, cognitive intervention, along with lifestyle modifications (including balance training, stretching, and walking), resulted in improvement in functions such as phonemic total, memory delayed recall, and quality of life. Furthermore, anxiety and depression were observed to be significantly reduced for the same group [60].

Cognitive training program (ACTIVE): Another study reported that the use of ACTIVE cognitive training, which is for 12 weeks, was noted to have improved cognitive functions and that the result even lasted for a minimum of 9 months [41].

Omega 3 polyunsaturated fatty acid supplementation and a multidomain intervention: Effect of omega 3 polyunsaturated fatty acid supplementation and a multidomain intervention (physical activity, cognitive training, and nutritional advice), alone or in combination, compared with placebo, on cognitive decline was studied (The Multidomain Alzheimer Preventive Trial, MAPT). No significant effects on cognitive decline over 3 years in elderly people with memory complaints was seen. In their study, it has been suggested to create an effective multidomain intervention strategy to prevent or delay cognitive impairment in the target population, particularly in real-world settings [65].

Cognitive intervention and lifestyle modifications: Interventions utilized a pencil-and-paper method to stimulate cognitive functions (like memory, executive, attention, visuospatial and language functions), physical exercises (including balance training, stretching, and walking), and lifestyle modification, along with home-based lifestyle modifications bordering on regular exercises, low alcohol consumption, smoking cessation, increasing cognitive activities, increasing social activities, and consuming a good diet for brain function, as well as controlling hyperlipidemia and hypertension. The subjects in both groups also received education on the prevention and risk factors of dementia and also on how to go about making lifestyle modifications. Functions such as phonemic total, memory delayed recall, and quality of life were significantly improved for the group that had cognitive intervention along with exercises and lifestyle modifications. Furthermore, anxiety and depression were observed to be significantly reduced for the same group [60].

Cognitive, functional, and psychoeducation program: The Rehacop is a cognitive rehabilitation program, theoretically based on strategies of rehabilitation (restoration, compensation, and optimization). The Rehacop uses a bottom-up approach. Top-down training with the ADL module is used later to help with the generalization of gains to the participant’s life. It is a 5-month intervention allowing either an individual or group approach. The results showed direct effects on neurocognition, as well as far-transfer effects on apathy, QoL, and subjective complaints after the intervention. The effect size was small for neurocognition, medium for apathy and QoL, and large for participants’ subjective complaints [66].

The effect of multicomponent nonpharmacological interventions on the quality of life of subjects with SCD was directly mentioned in three studies [59,60,66]. It is impossible to leave out how the significant reduction in anxiety, depression, and perceived stress as reported will contribute to overall wellbeing, which could translate to improved quality of life [58,62,63]. Another critical aspect that seems to be scarcely addressed when selecting multicomponent nonpharmacological interventions has to do with the length of time used in giving the intervention. According to a study, the increase in the use of management strategies is often the most significant predictor for subject memory complaint improvement [61]. This could mean that the more an SCD subject engages an effective intervention, the higher the prospect of actualizing a positive result.

This study has tried to collect all up-to-date evidence of non-pharmacological multi-component interventions for SCD in older adults. The amount of available information is insufficient to draw firm conclusions regarding the type and combination of multi-component interventions to be used for SCD in older adults. Larger trials utilizing multi-component interventions may be conducted for concrete results on the type of interventions beneficial for preventing or delaying cognitive decline. Most importantly, the development of an assessment scale for SCD is much required. Future studies could assess the extent to which intervention characteristics, such as the number, length, and frequency of sessions, and patient characteristics (such as demographics, education. daily functioning, and social activity), have an impact on the outcomes.

## 5. Conclusions

Subjective cognitive decline remains a concern that individuals will always look to beat as they near the latter part of their existence, and based on the findings from this review, it is apparent that a much more significant result can be attained by adopting multicomponent interventions. However, while different studies have shown promises of not only helping people reduce the reportage of SCD but also improving cognitive performances or delaying cognitive decline, at least there is still more work to be done in order to arrive at multicomponent nonpharmacological interventions that are widely accepted and highly effective.

## Figures and Tables

**Figure 1 geriatrics-08-00004-f001:**
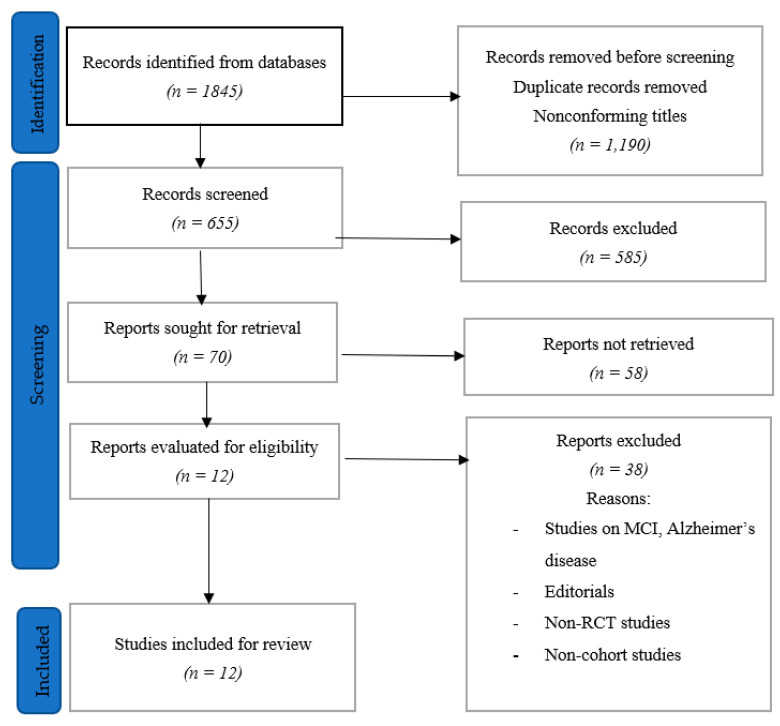
Study selection process.

**Table 1 geriatrics-08-00004-t001:** Summary of study characteristics.

Author	Study Design	No. of Subjects	Age: Mean(±S.D)/Range (Years)	Intervention & Description	Duration	Outcomes
Boa Sorte Silva et al. (2018) [57]	Randomized controlled trial	127	67.5 (±7.3)	- Intervention Group (IG): Multiple-modality (involving a 5 min warm-up; 20 min aerobic exercise; 5 min cool down; 10 min resistance; 5 min stretching) and 15 min mind-motor training (square-stepping exercise) - Control Group: Multiple-modality intervention + 15 min of balance, range of motion, and breathing exercise.	24 weeks (3 days/week)	No significant differences in the mean change when the global cognitive functioning (GCF) and memory scores were compared with baseline observation. The intervention group, however, had significant improvements in memory and GCF compared to the control group at 24 weeks and 52 weeks (follow-up). Both groups showed slight improvements in concentration, reasoning, and planning.
Montoya-Murillo et al. (2020) [66]	Randomized controlled trial	124	55 and above	Rohacop: A comprehensive interactive intervention that entails sessions on attention and concentration for over 4 weeks; learning and memory for 3 weeks; language for 3 weeks; executive functioning for 3 weeks; and processing speed training given throughout. Control: Occupational tasks including singing, drawing, crafts, gardening, and reading and commenting on a newspaper.	3 months	The participants in the Rohacop group showed significant improvement in neurocognition compared to the control group. The mini-mental state examination (MMSE) of the Rohacop group was higher than the control group, albeit not significantly. Again, there was a significant reduction in subjective complaints for both groups.
Zuniga et al. (2016) [58]	Randomized controlled trial	179	66.4 (±5.7)	- Aerobic exercise group: Walking/brisk walking for 10–40 min; warm-up for 5–10 min; and cool-down stretches. - Resistance training group: 4 muscle-toning exercises (using resistance bands or dumbbells); 2 balance exercises; 1 yoga sequence; and 1 other preferred exercise. - The exercises were done for 1 h thrice a week.	12 months	No significant difference was found in subject memory impairment for both groups, though SCD was observed to be stable throughout. The interventions, however, brought about significant effects on happiness, perceived stress reduction, and SCD reporting. This led to the conclusion that SCD is more of a psychological event than a neurological one.
Frankenmolen et al. (2018) [61]	Randomized controlled trial	60	66.2 (±7.2)	- Memory strategy training (MST): Involves 7 sessions bordering on education about memory, formulating memory goals, engaging in cognitive restructuring, taking notes, putting objects in conscious and fixed places, and paying attention, among other activities. - Control: Computer program revolving around attention and memory tasks. Also involved 7 sessions.		No significant improvement in subjective memory impairment after the interventions were given, but 4% of the subjects in each group improved clinically and significantly in terms of subjective memory complaints. The increase in the use of strategies was found to be the strongest predictor for SMC improvement in the MST group. Larger significant improvement in personal memory goals was also found for the MST group. Overall, significant improvements in 2 memory tests—memory test performance and subjective memory functioning in daily life—were observed in both groups.
Ramnath et al. (2021) [59]	Quasi-experimental	45	72 (±5)	- Interactive video games (IVG): Six games (including ten-pin bowling, boxing, track and field, table tennis, beach volleyball, and soccer) played by subjects for 30 min daily. - Conventional multimodal intervention (CMI): Included 10 min warm-up; 30 min strength training; 10 min proprioceptive exercises, and 10 min of cooling off	12 weeks	There were significant improvements in cognitive measures of processing speed and executive functioning in the IVG group compared to the CMI group. Furthermore, global cognitive functioning was found to be significantly improved for the IVG group. A significantly better improvement in balance and mobility was also realized with the IVG group, and this indirectly meant improvement in quality of life.
Cohen-Mansfield et al. (2015) [62]	Randomized controlled trial	44	65 and above	Educative courses leading to the development of the following: - Health Promotion Course (HPC): Involves teaching topics like health behaviors; dementia, and delirium; communication; cognitive activities to keep the mind fit; relationships; depression; life-long learning, etc. - Cognitive Training Course (CTC): Hinges on Advanced Cognitive Training for Independent and Vital Elderly (ACTIVE) course. It focuses on memory, reasoning, processing speed, but only the memory training aspect was employed in this study. - Participation-centered Course: Focuses on memory, cognitive, and organizational strategies. All the subjects underwent a goal attainment process.	10 weeks	All interventions showed significant improvements in global cognitive function score compared to what was obtainable for the baseline. However, the CTC group was observed to have significantly better global cognition score than the other groups. Slight improvements in executive function, visual spatial function, and attention were found across all the groups. Plus, loneliness and self-report on memory difficulties also declined with all the interventions.
Oh et al. (2017) [64]	Randomized controlled trial	53	50–68	- SMART (Smartphone-based brain Anti-ageing and memory Reinforcement Training): Focuses on attention, memory, and working memory, with subjects being trained for 15–20 min for 5 days a week. - Fit Brains: Focuses on concentration, logic, processing speed, visual. All the activities can be completed in 2 min, and the subjects did them twice for 5 days per week.	8 weeks	Working memory scores were found to be significantly improved in the SMART group than the Fit Brains group. Subjective memory complaints, however, only declined in the Fit Brains group.
kwok et al. (2012) [41]	Randomized placebo-controlled trial	223	65 and above	- Cognitive training program (ACTIVE), dwelling on attention, memory, and reasoning. The intervention involves activities like playing games, mnemonic strategies, and goal identification. - Control: health-related educational lectures. - Cognitive training program (ACTIVE), dwelling on attention, memory, and reasoning. The intervention involves activities like playing games, mnemonic strategies, and goal identification. - Control: health-related educational lectures.		Sognitive functions concerning memory and conceptualization improved significantly, and the outcome was sustained for 9 or more months.
Andrieu et al. (2020) [65]	Multicenter RCT	1680	70 and above	- lifestyle intervention including cognitive training and counseling on physical activity and nutrition - Lifestyle intervention plus omega-3 fatty acid supplementation - Omega-3 fatty acid alone	3 years	Slight improvements in cognitive performance, although these were not statistically significant across all groups.
Chan et al. (2017) [25]		56	60 and above	- Dejian Mind–Body Intervention (DMBI): Involves Chan practice (whereby subjects were guided on self-awareness and self-control in relation to unrealistic goals); Nei Gong practice (that has to do with the teaching of mind–body exercises bordering on a series of breathing exercises, and calm and gentle movements); and dietary modifications. Conventional Memory Intervention (CMI): Involves psychoeducation, mnemonic training, and progress monitoring	10 weeks	Total learning and recognition performances significantly improved for both groups. The DMBI group showed significant improvement in immediate recall and delayed recall when visual memory was considered. Furthermore, the CMI group showed significant improvement in delayed recall. Subjective feelings of the subjects as per their physical and psychological health also improved significantly in the DMBI group, unlike the CMI group.
Pereira-Morales et al. (2017) [63]	Randomized controlled trial	40	66.4 (±5.6)	- Integrated Psychostimulation Program (IPP): A combination of computerized cognitive training (CCT) and conventional cognitive training (like meta-memory activities, progressive relaxation exercises, group discussions, and pen and paper exercises. Done for 90 min for 4 days per week. - Computerized cognitive training: Training delivered through a web application for 60 min for 4 days per week	8 weeks	Subjective memory complaints declined significantly for the IPP group. Furthermore, significant improvements were observed for the IPP group regarding short-term memory, long-term memory, processing speed, and phonological verbal fluency. Symptoms of anxiety also declined significantly for the IPP group. The CCT alone group, however, showed significant improvement in cognition and short-term memory.
Hong et al. (2020) [60]	Prospective controlled trial	56	55–75	- Cognitive intervention + lifestyle modifications: involving a pencil-and-paper method to stimulate cognitive functions (like memory, executive, attention, visuospatial and language functions); 10 min of physical exercises (including balance training, stretching, and walking) and lifestyle modification. The cognitive intervention went on for 90 min per session for 2 days every week. - Home-based lifestyle modifications bordering on regular exercises, low alcohol consumption, smoking cessation, increasing cognitive activities, increasing social activities, and consuming a good diet for brain function, as well as controlling hyperlipidemia and hypertension. The subjects in both groups also received education on the prevention and risk factors of dementia and also on how to go about making lifestyle modifications.	12 weeks	There were no significant improvements recorded when the cognitive function—as reflected by the CANTAB scores—was measured across the two groups. However, functions, such as phonemic total, memory delayed recall, and quality of life, were significantly improved for the group that had cognitive intervention, along with exercises and lifestyle modifications. Furthermore, anxiety and depression were observed to be significantly reduced for the same group.

## Data Availability

Not applicable.

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
