# Peer review of "Multi-Component Interventions in Older Adults Having Subjective Cognitive Decline (SCD)—A Review Article"

_geriatrics, 2022, doi:10.3390/geriatrics8010004_

Round 1

Reviewer 1 Report

Although the paper has several positive qualities, there are some major concerns, on the basis of which I recommend to reject the paper. The authors perform a ¿systematic / literature? review of multicomponent interventions in SCD in older adults. This is an article on an important topic, however, the overlap with the meta-analysis of Bhome et al. (2018) is too large to make a significant contribution to the literature.

Some minor concerns:

- Conceptual flaws in the introduction.

- Detailing in more depth both the possible causes of SCD and non-pharmacological interventions.

- The format for citing studies follows both APA and Vancouver standards, it would be necessary to follow the format established by the journal.

Some major concerns:

- There is no prospero protocol.

- Review of the keywords to be included: specify why some of them have been included, such as spirituality, and nutrient, which I believe do not respond to the and objective of the r…w as it is proposed or perhaps need to justify their inclusion.

- The type of studies included: RCTs, cohort studies, reviews, systematic reviews.... It is a very wide range not specified the reason to do so.

- The authors themselves state that there is a meta-analysis that responds to their objective and was published in 2018. therefore, why is this review not updated with data from 2019-2022. They do not see the need to include earlier studies already collected in the meta-analysis of Bhome et al.

Author Response

Response to Reviewer 1 Comments

Point 1: Although the paper has several positive qualities, there are some major concerns, on the basis of which I recommend to reject the paper. The authors perform a ¿systematic / literature? review of multicomponent interventions in SCD in older adults. This is an article on an important topic, however, the overlap with the meta-analysis of Bhome et al. (2018) is too large to make a significant contribution to the literature.

Response 1:

The authors performed a literature review for multicomponent non-pharmacological interventions in SCD in older adults. This study is different from the study by Bhome et al (2018) and not an overlap.

  Interventions categorised by Bhome et al (2018) are Psychological, Cognitive, Lifestyle and Pharmacological. In Bhome et al’s study, out of 20 included studies, only two studies with regards to lifestyle interventions for SCD involving physical activities/ exercises was evaluated. In one study, both groups (aerobic exrcise and gentle exercise) showed significant improvement in objective cognitive performance. Another study, did not show significant improvement and stated limited evidence available with regards to lifestyle interventions for SCD (Page 8).

  To tackle this limitation, our study categorises all non-pharmacological interventions such as, Physical activity, Pychosocial education, Computer program, Lifestyle modifications, and Dietary measures. Researches in last decade have emphasized on the role of Physical activity to maintain or prevent cognitive decline. In this study we have not included pharmacological intervention as seen in study by Bhome et al. Thus, our study has tried to gather all inputs regarding multicomponent non-pharmacological intervention in SCD in older adults. Hence this study is not an overlap.

Point 2: Some major concerns:

- There is no prospero protocol.

- Review of the keywords to be included: specify why some of them have been included, such as spirituality, and nutrient, which I believe do not respond to the and objective of the r…w as it is proposed or perhaps need to justify their inclusion.

- The type of studies included: RCTs, cohort studies, reviews, systematic reviews.... It is a very wide range not specified the reason to do so.

- The authors themselves state that there is a meta-analysis that responds to their objective and was published in 2018. therefore, why is this review not updated with data from 2019-2022. They do not see the need to include earlier studies already collected in the meta-analysis of Bhome et al.

Response 2:

  • As our study is a Literature Review, Prospero registration is not required.
  • Key words appropriate for all non-pharmacological interventions have been included. ‘Spirituality’ is taken as a key word as it is associated with meditation, e.g., study by Marciniak et al (2014) and many related studies have shown its beneficial effects on cognitive health. Keyword ‘nutrient’ is taken, because many evidence based- studies confirm that dietary and nutrition patterns have a positive effect on cognitive performance, e.g., Klimova et al (2020).

  • In our study we have included a wide range of studies but limited the years of publication between 2012 – 2022, a timeframe of 10 years. This avails the researcher to explore maximum data that are more current for the subject matter.

  • As mentioned earlier in Response 1, interventions categorized by Bhome et al (2018) are psychological, cognitive, lifestyle and pharmacological. In our study, we have included physical activity and focused only on nonpharmacological multi component intervention for SCD in older adults. Hence it was necessary to do a review of last ten years studies to explore maximum literature available.

Point 3: Some minor concerns:

- Conceptual flaws in the introduction.

- Detailing in more depth both the possible causes of SCD and non-pharmacological interventions.

- The format for citing studies follows both APA and Vancouver standards, it would be necessary to follow the format established by the journal.

Response 3:

  • Changes has been done in introduction, as suggested.
  • A detail discussion on possible causes of SCD and categorising non-pharmacological interventions has been done.
  • The format for citing studies has been corrected to Vancouver standards.

Reviewer 2 Report

A Review of multi-component Interventions in older adults having Subjective cognitive decline (SCD)

The aim of the paper was a review of subjective cognitive impairment in older people, the relevance of which lies in the association with Alzheimer's disease. This paper shows ways to improve the event and suppress the rate of cognitive decline. It shows a compilation of articles from Pubmed, Scopus and EBSCO with relevant data on the subject.

It reflects a broad perspective on the subject, which is of great value to researchers in the field.

However, I would like to make some recommendations:

-Improve whether it is a systematic review or literature review, and incorporate that improvement in title, objectives, abstract, method..etc.

-Unify apa and Vancouver format.

-It is very important to mention the practice of physical activity in the introductory part in such a relevant area of quality of life in older adults:

"The practice of physical activity is one of the variables that should not be forgotten in this area, as it means an improvement in the quality of life of older people as well as a guarantee of good ageing in terms of health https://doi.org/10.3390/bs12090331 “.  The practice of physical activity should be taken into account as a measure of health protection and functional skills that result in a better quality of life for older people, which results in better physical health for older people https://doi.org/10.3390/socsci11060265". Likewise, socio-environmental factors related to a higher quality of life also influence a greater practice of physical activity in older people, such as income and education, being education on healthy lifestyle habits one of the axes to acquire healthy lifestyle habits, such as practising more physical activity in older people ​​https://doi.org/10.3390/ijerph182010815".

At the end of your discussion I suggest:

-Include: Practical implications of research on older people.

-Include: Theoretical implications for other researchers or academics.

-Include: Limitations.

-Include: strengths of your study in comparison to others.

-Include: Future lines of research.

Congratulations on the study and welcome these positive suggestions in order to improve your internationalisation, downloads and citations.

Author Response

Point 1: Improve whether it is a systematic review or literature review, and incorporate that improvement in title, objectives, abstract, method..etc.

Response 1: Our study is a literature review. The change in title has been incorporated. “Multi-component Interventions in older adults having Subjective cognitive decline (SCD) – A Review article”. The changes as suggested in title, abstract, objective and method are done.

Point 2: Unify apa and Vancouver format.

Response 2: The format for citing studies has been corrected to Vancouver standards.

Point 3: It is very important to mention the practice of physical activity in the introductory part in such a relevant area of quality of life in older adults:

"The practice of physical activity is one of the variables that should not be forgotten in this area, as it means an improvement in the quality of life of older people as well as a guarantee of good ageing in terms of health https://doi.org/10.3390/bs12090331 “.  The practice of physical activity should be taken into account as a measure of health protection and functional skills that result in a better quality of life for older people, which results in better physical health for older people https://doi.org/10.3390/socsci11060265". Likewise, socio-environmental factors related to a higher quality of life also influence a greater practice of physical activity in older people, such as income and education, being education on healthy lifestyle habits one of the axes to acquire healthy lifestyle habits, such as practising more physical activity in older people ​​https://doi.org/10.3390/ijerph182010815".

Response 3: Effect of Physical activity on cognition has been elaborated in Introduction part (management of SCD) and Discussion as suggested.

Point 4: At the end of your discussion I suggest:

-Include: Practical implications of research on older people.

-Include: Theoretical implications for other researchers or academics.

-Include: Limitations.

-Include: strengths of your study in comparison to others.

-Include: Future lines of research.

 Response 4: Theoretical and practical implications of the study are included along with objective and in discussion. Strength, limitations and future lines of research has been included at end of discussion.

Round 2

Reviewer 1 Report

Dear authors,

Thank you for your effort to improve the manuscript. The manuscript now is clear, relevant for the field and is presented in a well structured manner.

The discussion and conclusion are consistent with the evidence found and now it is well discussed and presented.

Now I think this manuscript provides an advancement of the current knowledge.